# Mind Your Step: Continuous Conditional GANs with Generator Regularization

**Yunkai Zhang**[*]
Department of IEOR
University of California, Berkeley
Berkeley, CA 94708
yunkai_zhang@berkeley.edu

**Yufeng Zheng**[*]
Department of IEOR
University of California, Berkeley
Berkeley, CA 94708
yufeng_zheng@berkeley.edu

**Xueying Ma**[†]
Department of IEOR
Columbia University
xm2209@columbia.edu

**Siyuan Teng**
Department of IEOR
University of California, Berkeley
siyuan_teng@berkeley.edu

**Jingshen Wang**
Division of Biostatistics
University of California, Berkeley
jingshenwang@berkeley.edu

**Zeyu Zheng**
Department of IEOR
University of California, Berkeley
Berkeley, CA 94708
zyzheng@berkeley.edu

## Abstract

Conditional Generative Adversarial Networks are known to be difficult to train, especially when the conditions are continuous and high-dimensional. To partially alleviate this difficulty, we propose a simple generator regularization term on the GAN generator loss in the form of a Lipschitz penalty. The intuition of this Lipschitz penalty is that, when the generator is fed with neighboring conditions in the continuous space, the regularization term will leverage the neighbor information and push the generator to generate samples that have similar conditional distributions for neighboring conditions. We analyze the effect of the proposed regularization term and demonstrate its robust performance on a range of synthetic tasks as well as real-world conditional time series generation tasks.

## 1 Introduction

Conditional Generative Adversarial Networks (cGANs) [1] are a powerful class of generative models where the goal is to learn a mapping from input to output distributions conditioned on some auxiliary information, such as class labels [1, 2], images [3, 4], or text [5, 6]. While cGANs have demonstrated outstanding capabilities in a wide range of conditional generation tasks, they are also known to be difficult to train since the optimization objective must consider various input conditions and is cast as a min-max game between the generator network and the discriminator network. Much past work has been devoted to stabilizing the training of GANs. For example, [7] introduces Wasserstein-GAN (WGAN) that uses the Earth Mover distance as a more explicit measure of the distribution divergence in the loss function.

---

[*]The authors contribute equally.
[†]Currently works at Apple.

NeurIPS 2022 Workshop on Synthetic Data for Empowering ML Research.

To better enforce the $k$-Lipschitz assumption in WGANs, [8] presents a regularization term on the discriminator. [9] studies the issue of mode-collapse, where only a small subset of the true output distribution is learned by the generator [10], by encouraging the generator to produce diverse outputs based on the latent input noise. On the other hand, [11] proposes to penalize the discriminator from being overly sensitive to small perturbations to the inputs through consistency regularization by augmenting the data passed into the discriminator during training.

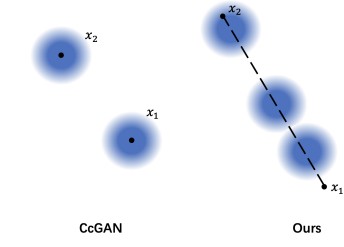

Figure 1: CcGAN only take small steps around sampled conditions, while we generalize better by also regularizing on interpolated conditions.

However, new challenges arise when the given conditions are continuous and multi-dimensional, which are often observed in real-life scenarios. One example is to generate spatial distributions of taxi's drop-off locations conditioned on its pick-up time and locations [12], or to synthesize facial images conditioned on age [13]. Among them, time series generation is not only an especially challenging task due to the complex temporal dynamics in the conditions, but also a tool with enormous social impact in applications such as healthcare, where patients' data cannot be used directly due to privacy concerns. In these practical domains, it is highly likely that not every possible condition can be represented in the training data, which we denote as *gaps*, and the neural network generator might extend poorly to those unseen conditions. To address such concerns, [13] introduces Continuous Conditional GAN (CcGAN) and suggests to add Gaussian noises to each sample of the input conditions in order to cover the gaps, at the cost of less sensitivity of the generator to more granular changes in the input conditions. In light of these observations, we propose a simple but effective generator regularization term on the Conditional GAN generator loss in the form of Lipschitz penalty. The intuition is that our model should only take small steps when a small perturbation is applied to any condition not only in the training set, but also on the interpolation between samples in the training set. Figure 1 shows a visual comparison with CcGAN. In summary, our contributions are three-fold:

- Synthetic experiments reveal that CcGAN and vanilla cGANs could behave sub-optimally when the dimension of conditions or the number of gaps in the training set increases.

- We propose a regularization approach that encourages the generator to leverage neighboring conditions in the continuous space through Lipschitz regularization without sacrificing the generator's faithfulness to the input conditions.

- We formulate the conditional time series generation problem and show the effectiveness of the proposed method, especially when generalizing to unseen conditions.

## 2 Method

**Challenges of Continuous, Multi-Dimensional Conditions.** cGAN commonly suffers from two problems, especially when the conditions are supported in a multi-dimensional and continuous space. **(P1)** Since the condition space $\mathcal{X}$ is continuous and multi-dimensional, $\boldsymbol{x}_i$'s in the training set are very likely to be different from each other. To make it worse, at each forward propagation only one noise $\boldsymbol{z}_j$ is sampled from $\mathcal{Z} \sim p_z(\boldsymbol{z})$, which is common in the practice training of cGAN. Therefore, for a certain $\boldsymbol{x}_i$, the discriminator can only get the information of $G(\boldsymbol{x}_i, \boldsymbol{z}_j)$ and may find it particularly challenging to generalize to the general distribution of $G(\boldsymbol{x}_i, \mathcal{Z})$. **(P2)** As we increase the number of dimensions for $\mathcal{X}$, the conditions observed $\{\boldsymbol{x}_i\}_{i=1}^N$ become more sparse and more gaps are created. For most conditions $\boldsymbol{x} \in \mathcal{X}$, few or even no samples can be observed during training. In most cGAN literature, when training cGANs, the generator are only given and trained on the conditions observed in the training set. As a result, the generator might extend poorly when given a new condition from $\mathcal{X}$ that has never been observed in the training set.

**GR-cGAN.** To address the aforementioned issues, we propose a novel regularization of the generator and name the resulting model as **Generator Regularized-cGAN (GR-cGAN)**. We first present the expression of the regularization term, and then discuss how it can remedy these problems.

The generator regularization is based on a continuity assumption of the conditional distribution $p_r(\boldsymbol{y}|\boldsymbol{x})$, where $\boldsymbol{y}$ denotes the dependent term. For a wide range of applications, but not all, it is

natural to assume that a minor perturbation to the condition $\boldsymbol{x}$ will only slightly disturb the conditional distribution $p_r(\boldsymbol{y}|\boldsymbol{x})$. On a high level, we hope that the distribution of $G(\boldsymbol{x}, \boldsymbol{z})$, which is used to approximate the distribution of $p_r(\boldsymbol{y}|\boldsymbol{x})$, shifts smoothly as we change $\boldsymbol{x}$. Since directly regularizing the generator from a distribution perspective can be challenging, we instead regularize the gradient of $G(\boldsymbol{x}, \boldsymbol{z})$ with respect to $\boldsymbol{x}$. Specifically, we add the following regularization term to the generator loss, to encourage the optimized generator $G$ to obtain a smaller loss on this regularization term:

$$\mathcal{L}_{GR}(G) = \mathbb{E}_{\substack{\boldsymbol{z} \sim q(\boldsymbol{z}), \\ \boldsymbol{x} \sim \tilde{p}(\boldsymbol{x})}} ||\nabla_{\boldsymbol{x}} G(\boldsymbol{x}, \boldsymbol{z})||, \tag{1}$$

where $\nabla_{\boldsymbol{x}} G(\boldsymbol{x}, \boldsymbol{z})$ is the Jacobian matrix. The empirical distribution $\tilde{p}(\boldsymbol{x})$ indicates the locations where we regularize the Jacobian matrix $\nabla_{\boldsymbol{x}} G(\boldsymbol{x}, \boldsymbol{z})$, and can be implicitly defined by sampling uniformly along straight lines between pairs of conditions sampled from the training set. This sampling method allows us not only to take small steps on the conditions observed in the training set, but also to take small steps on the conditions that have not been observed in the training set but might occur while testing or applying the trained cGAN generator. If the conditions fall in a high-dimensional space where linear interpolation is not feasible, we can first project the conditions onto another vector space - for example, onto the latent space of variational autoencoders (VAEs) - before interpolations [14, 15]. See Figure 4 for details of applying GR-cGAN to conditional time series generation. Intuitively, when $\mathcal{L}_{GR}(G)$ takes a small value, for any fixed $\boldsymbol{z} = \boldsymbol{z}_0$, the output of the generator $G(\boldsymbol{x}, \boldsymbol{z}_0)$ will only shift moderately and continuously as $\boldsymbol{x}$ changes.

Finally, the cGAN objective with generator regularization now becomes

$$\min_{G} \max_{D} \quad L(D, G)$$
$$:= \mathbb{E}_{(\boldsymbol{x}, \boldsymbol{y}) \sim \hat{p}(\boldsymbol{x}, \boldsymbol{y})}[\log D(\boldsymbol{x}, \boldsymbol{y})] + \mathbb{E}_{\boldsymbol{z} \sim p_z(\boldsymbol{z}), \boldsymbol{x} \sim \hat{p}(\boldsymbol{x})}[\log(1 - D(G(\boldsymbol{x}, \boldsymbol{z})))] + \lambda \cdot \mathcal{L}_{GR}(G).$$

The term $\lambda$ controls the degree of regularization. In other words, a larger $\lambda$ discourages the model from reacting drastically to small perturbations in the input conditions.

### How does generator regularization overcome (P1) and (P2)?

For (P1), during training, a batch of $(\boldsymbol{x}_i, \boldsymbol{y}_i)$ pairs are sampled from the training set. For any $\boldsymbol{x}_i$ from this batch of data, when the generator regularization is applied, the samples in the vicinity of $\boldsymbol{x}_i$ are encouraged to facilitate the training of the generator and the discriminator. In the case where the generated distribution of $G(\boldsymbol{x}_i, \boldsymbol{z})$ with $\boldsymbol{z} \sim p_z(\boldsymbol{z})$ is concentrated on a pathological mode collapse distribution (in other words, the generator always underestimate the variance of the ground-truth conditional distribution $p_r(\boldsymbol{y} \mid \boldsymbol{x}_i)$), the discriminator can better detect local mode collapse and learn to classify such pathological distribution as fake, thus improving the generator in return.

For (P2), when given a new condition $\boldsymbol{x}_0$ that does not exist in the train set, the conditional distribution given by the generator in GR-cGAN on $\boldsymbol{x}_0$ is similar to the conditional distribution given on the conditions in the vicinity of $\boldsymbol{x}_0$ in the training set. If we penalize the gradient from being too large, we are effectively encouraging the model to learn a smooth transition between each pair of samples from the training set and thus generalize to close these gaps.

We compare the proposed GR-cGAN with related works in Appendix B.

## 3 Analysis of the Proposed Regularization

Following the definition of Lipschitz continuity for functions, we first deliver a formal definition of continuous conditional distribution named **Lipschitz continuous conditional distribution**. Next, we present its connection with the proposed generator regularization.

**Definition 3.1 ($K$-Lipschitz Continuous Conditional Distribution)** *Let $X$ and $Y$ be random variables with support $R_X$ and $R_Y$ respectively. Denote the distribution induced by $X \mid Y = y$ as $\mathcal{F}_y$. We say $X$ has a $K$-Lipschitz continuous conditional distribution with respect to $Y$, if for all $y_1, y_2 \in R_Y$, the Wasserstein distance between $\mathcal{F}_{y_1}$ and $\mathcal{F}_{y_2}$ satisfies*

$$W(\mathcal{F}_{y_1}, \mathcal{F}_{y_2}) \leq K \cdot \|y_1 - y_2\|,$$

*where $W(\cdot, \cdot)$ denotes the Wasserstein distance between two distributions, and $|| \cdot ||$ indicates a norm.*

Note that when the Wasserstein distance is used to evaluate the distance between two probability distributions, cGANs can be extended to conditional Wasserstein GANs. Other distance measures to quantify the distance between two conditional distributions can also be adapted.

Given two arbitrary conditions $x_1$ and $x_2$, suppose that the generator satisfies $||G(x_1, z_0) - G(x_2, z_0)|| \leq K_0 \cdot ||x_1 - x_2||$ for any $z_0$. The conditional distribution given by the generator on $x_1$ and $x_2$ are $G(x_1, z)$ and $G(x_2, z)$ with $z \sim p_z(z)$ respectively. Notice that when the generator regularization is applied, the term $K_0$ will be pushed to a smaller level. We prove in Appendix C that $W(G(x_1, z), G(x_2, z)) \leq K_0 \cdot ||x_1 - x_2||$, which indicates that the conditional distribution learned by the generator is a $K_0$-Lipschitz continuous conditional distribution with respect to $x$. With the use of generator regularization, the conditional distribution given by the generator is encouraged to be more continuous from the perspective of K-Lipschitz continuous conditional distributions.

## 4 Experiments

To demonstrate the effectiveness of the proposed generator regularization term, we begin our experiments with a simple 2D synthetic dataset and then proceed with a real-world conditional time series generation task. Our code is here[3].

### 4.1 Circular 2-D Gaussians

We generate a synthetic dataset from 2-D Gaussians with different means following the experiments in CcGAN [13]. The condition $x$ has a dimension of one which measures the polar angle of a given data point and the dependency $y$ has a dimension of two. Given $x \in [0, 2\pi]$, we construct $y$ such that it follows a 2-D Gaussian distribution, specifically,

$$y \sim \mathcal{N}(\mu_x, \Sigma) \text{ with } \mu_x = \left( \begin{array}{c} R \cdot \sin(x) \\ R \cdot \cos(x) \end{array} \right), \text{ and } \Sigma = \tilde{\sigma}^2 I_{2 \times 2} = \left( \begin{array}{cc} \tilde{\sigma}^2 & 0 \\ 0 & \tilde{\sigma}^2 \end{array} \right).$$

For a thorough analysis, we study several different settings for $x$ when generating the dataset. In Section 4.1.1, we choose a subset of $[0, 2\pi]$ for training and evaluate how well the models can generalize to the gaps that are absent during training. In Appendix E.1.1, we further report the experiment results when $x$ is evenly distributed in the range of $[0, 2\pi]$.

### 4.1.1 Partial Dataset

To examine the robustness of each model to the presence of gaps in the training set, we intentionally select a subset of $[0, 2\pi]$ and only train the models on the subset. Specifically, we set three gaps with a length of $\pi/12$, and remove these gaps from the range $[0, 2\pi]$ to get a subset of $[0, 2\pi]$. These three gaps are non-overlapping and are evenly located in $[0, 2\pi]$. We set $x$ to 120 different values that are evenly arranged in the subset, which are then used as the train labels.

Each value of $x$ is the mean of a Gaussian distribution. For each gap, we use the angle in the middle of the gap as the test label to evaluate the performance of the models. Thus, the three gaps correspond to three test labels. The positions of the train labels and test labels are shown in Figure 2(a). To generate a training set, for each $x$ in the train labels, 10 samples are generated. We denote this training set as a partial dataset. For $R$ and $\tilde{\sigma}^2$, we set $R = 1$ and $\tilde{\sigma}^2 = 0.2$. Figure 2(b) shows 1,200 training samples on the partial dataset.

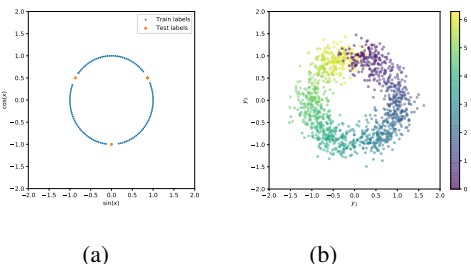

(a)      (b)

**Results:** The CcGAN (HVDL) and CcGAN (SVDL) models in CcGAN are considered as baseline models. We also consider the degenerated case of the proposed GR-cGAN (Deg.GR-cGAN) by setting the generator regularization coefficient $\lambda$ to

Figure 2: (a) illustrates the train labels and test labels. Given a label $x$, we plot a dot at $(\sin(x), \cos(x))$. The blue dots correspond to the train labels, while the orange dots correspond to the test labels. (b) gives the 1,200 samples in the training set. The color of each dot represents which train labels it belongs to.

---

[3]https://github.com/GR-cGAN/GR-cGAN.

zero. For the GR-cGAN model, we use the loss term given in Equation 1 where $\lambda = 0.02$. We generate 100 fake samples for each test label. We plot these fake samples in Figure 3. For each test label $x$, a circle that covers about 90% of the volume inside the pdf of $\mathcal{N}(\boldsymbol{\mu_x}, \boldsymbol{\Sigma})$ is also plotted. GR-cGAN achieves visually reasonable results. It is a good property that the generator can give reasonable fake samples even when given a label on a gap. GR-cGAN can be used in cases where there are missing labels in the training set. For example, in the task of generating photos of people with a given character description, if we only have samples of "young and happy" and "old and sad", we can use GR-cGAN to generate "old and happy" images.

More quantitative evaluation metrics are given in Appendix E.1.2, with unsurprising results that our GR-cGAN behaves better.

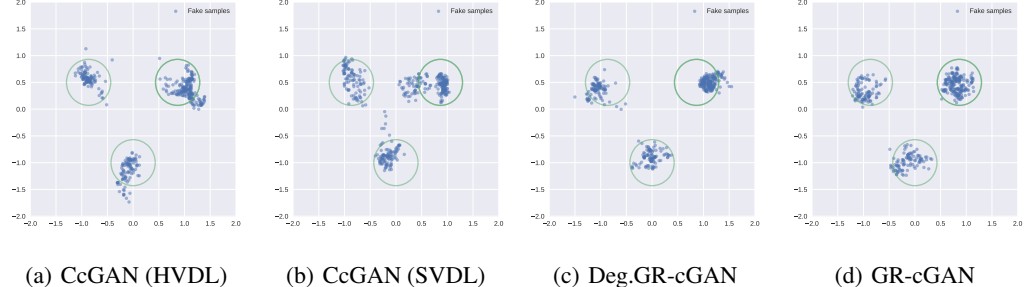

$\quad$ (a) CcGAN (HVDL) $\qquad$ (b) CcGAN (SVDL) $\qquad$ (c) Deg.GR-cGAN $\qquad$ (d) GR-cGAN

Figure 3: Visual results of the Circular 2-D Gaussians experiments on partial dataset. For each subfigure, we generate 100 fake samples using each GAN model at each of 3 labels in the test labels. The blue dots represent the fake samples. For each mean $x$ in the test labels, a circle that can cover about 90% of the volume inside the pdf of $\mathcal{N}(\boldsymbol{\mu_x}, \boldsymbol{\Sigma})$ is plotted.

## 4.2 Conditional Time Series Generation

In this section, we reformulate the settings defined in TimeGAN [16] to better study the effect of our generator penalty when the conditions are high-dimensional and more complex. Specifically, given multivariate sequential data of $T$ time steps $[y_i]_{i=1}^T$ and a distribution of latent noise $[z_{T+i}]_{i=1}^\tau$, our goal is to generate its evolution in the next $\tau$ time steps $p\left([y_{T+i}]_{i=1}^\tau \mid [y_i]_{i=0}^T\right)$. In our experiments, we always use the first one-third of the window as the condition, but can be easily extended to support conditions of arbitrary lengths.

**Model Architecture.** For a fair comparison, we avoid altering the original TimeGAN architecture unless noted and only make minimal changes to let TimeGAN support conditional generation. Here, we only highlight the differences we made. For more details on TimeGAN, we refer readers to [16]. As shown in Figure 8 in the Appendix, the generator $G$ is changed to an encoder-decoder network, where the conditions $[y_{0:T}]$ are first fed into the encoder. The decoder takes in the hidden states from the encoder as well as latent noise $[z_{0:\tau}]$. Due to the nature of high-dimensional time series, a simple linear interpolation of the conditions might not be feasible. Therefore, we first project the conditions into a latent space where an interpolation makes sense. A simple and readily available option is variational autoencoders (VAE). More details can be found in Appendix E.2.2. The VAE conditioning module is illustrated in Figure 4.

**Experiment Setup.** We adapt the stock[4] dataset from TimeGAN which consists of daily historical Google stocks data from 2004 to 2019. In addition, we also use the Electricity Transformer Temperature dataset[17] at a minute-level (ETTm1[5]) due to its more intricate and yet predicable dynamics. Data generated by different models are compared using two metrics - 1) **predictive score** ($\downarrow$): normalized-MAE where an off-the-shelf transformer forecasting model is trained on the generated dataset and then used to predict on the real dataset, and 2) **discriminative score** ($\downarrow$): a GRU

---

[4]https://github.com/jsyoon0823/TimeGAN/tree/master/data.

[5]https://github.com/zhouhaoyi/ETDataset/blob/main/ETT-small/ETTm1.csv.

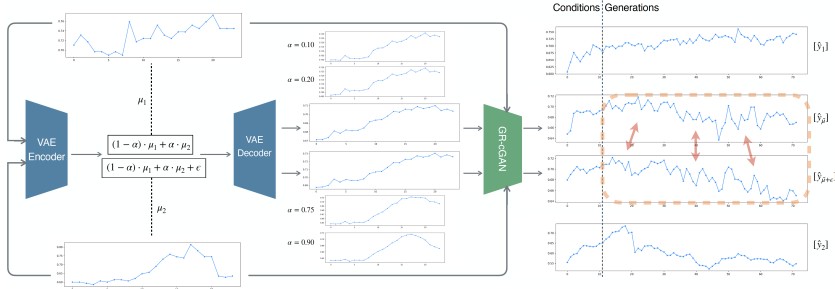

Figure 4: Interpolation between two conditions with a VAE module. The plotted windows are real model outputs on the "LULL" feature of ETTm1. Upon reconstructing the interpolated condition as well as its perturbed correspondence, we feed both into the generator and enforce the generated windows to be similar.

discriminator trained to classify whether a given sequence is generated or real, and the final score is the classification accuracy minus $0.5$. More details on how we tailored the evaluation protocol for conditional generation compared to TimeGAN can be found in Appendix E.2.1.

**Baselines.** **1) cTimeGAN** - our modified version of TimeGAN[16] so that it supports conditional inputs. **2) CcGAN**[13] - regularizes the generator by adding a simple perturbation on the conditions (i.e., $[y]_{i=0}^T \cdot (1 + \epsilon)$). **3) DGR-cTimeGAN** - an ablated version of our model where we do not penalize by perturbing the interpolation of two sampled conditions but instead perturb the two sampled conditions themselves.

| Dataset | #timesteps | CcGAN | | cTimeGAN | | DGR-cTimeGAN | | Ours | |
|---|---|---|---|---|---|---|---|---|---|
| | | Disc | Pred | Disc | Pred | Disc | Pred | Disc | Pred |
| Stock | 24 | **0.170** | **0.021** | 0.189 | 0.024 | **0.169** | **0.020** | 0.182 | **0.021** |
| ETTm1 | 72 | 0.390 | 0.060 | 0.412 | 0.063 | 0.367 | 0.057 | **0.363** | **0.056** |
| ETTm1$_{\text{all}}$ | 72 | 0.320 | 0.059 | 0.385 | 0.064 | 0.376 | 0.067 | **0.304** | **0.054** |

Table 1: Evaluation of time series generation using the first 1/3 of the window as conditions. *Disc* and *Pred* denote discriminative score and predictive score, respectively. Results for TimeGAN are averaged across 2 runs, while the results for our model are averaged across 5-7 runs. For CcGAN, we report the result from the best $\epsilon \in \{0.1, 0.01, 0.001\}$.

**Results.** Table 1 demonstrates that our model consistently outperforms other baselines except in terms of discriminative score on stock, even as we test a wide range of $\lambda$ as illustrated in Figure 5. In comparison, we show in Appendix E.3 that CcGAN is very sensitive to the choice of $\epsilon$. Additionally, cTimeGAN behaves poorly at inference time when a portion of the conditions is not used for training (i.e. when we manually enlarge the gaps in the training conditions), where on the ETTm1 dataset it reaches a high discriminative score of 0.412 compared to 0.363 of our model. On the other hand, our model achieves better scores when $30\%$ of the data is unseen during training compared to cTimeGAN trained using all data. The ablation results further demonstrate that regularizing the interpolation of conditions rather than the conditions themselves is critical for generalizing to new, unseen conditions at test time. It is worth noting that the reconstructed outputs of the VAE tends to be very smooth in our implementation as shown in Figure 4. We

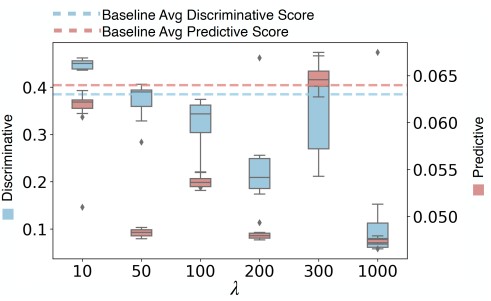

Figure 5: Sensitivity to regularization weight $\lambda$ on ETTm1$_{\text{all}}$. Horizontal dashed lines represent average cTimeGAN results. Lower scores indicate better performance.

expect our performance to improve further if an appropriate model is used to generate more realistic interpolation conditions for generator regularization.

## 5 Conclusion

In this work, we show a promising method to address the issues arisen in training conditional generative adversarial networks (cGANs) when the conditions are continuous and high-dimensional. We propose a simple generator regularization term on the GAN generator loss in the form of Lipschitz penalty. More specifically, the regularization term encourages the generator to take smaller steps when a small perturbation is applied to the conditions. Meanwhile, we acknowledge that a more sophisticated method is desired for interpolating a pair of conditions in terms of both visual quality and the ease of training.

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

## A  Problem Formulation

Let $\mathcal{X} \subset \mathbb{R}^m, \mathcal{Y} \subset \mathbb{R}^n, \mathcal{Z} \subset \mathbb{R}^l$ be the condition space, the output space, and the latent space respectively. Denote the underlying joint distribution for $\boldsymbol{x} \in \mathcal{X}$ and $\boldsymbol{y} \in \mathcal{Y}$ as $p_r(\boldsymbol{x}, \boldsymbol{y})$. Thus, the conditional distribution of $\boldsymbol{y}$ given $\boldsymbol{x}$ becomes $p_r(\boldsymbol{y}|\boldsymbol{x})$. The training set consists of $N$ observed $(\boldsymbol{x}, \boldsymbol{y})$ pairs, denoted as $\{(\boldsymbol{x}_i, \boldsymbol{y}_i)\}_{i=1}^N$. Following the vanilla cGAN [1], we introduce a random noise $\boldsymbol{z} \in \mathcal{Z}$ and $\boldsymbol{z} \sim p_z(\boldsymbol{z})$, where $p_z(\boldsymbol{z})$ is a predetermined easy-to-sample distribution. The goal is to train a conditional generator $G : \mathcal{X} \times \mathcal{Z} \to \mathcal{Y}$, whose inputs are the condition $\boldsymbol{x}$ and latent noise $\boldsymbol{z}$, in order to imitate the conditional distribution $p_r(\boldsymbol{y}|\boldsymbol{x})$. Our proposed regularization term is suitable for most variants of cGAN losses, such as the vanilla cGAN loss [1], the Wasserstein loss [7], and the hinge loss [18]. Without loss of generality, in Equation **??** we illustrate our regularization term on the vanilla cGAN loss. Below is the vanilla cGAN loss, where the conditional generator $G$ and discriminator $D$ are learned by jointly optimizing the following objective:

$$
\begin{aligned}
&\min_G \max_D \mathcal{L}_{cGAN}(D, G) \\
=&\mathbb{E}_{(\boldsymbol{x}, \boldsymbol{y}) \sim \hat{p}(\boldsymbol{x}, \boldsymbol{y})}[\log D(\boldsymbol{x}, \boldsymbol{y})] \\
&+ \mathbb{E}_{\boldsymbol{z} \sim p_z(\boldsymbol{z}), \boldsymbol{x} \sim \hat{p}(\boldsymbol{x})}[\log(1 - D(\boldsymbol{x}, G(\boldsymbol{x}, \boldsymbol{z})))],
\end{aligned}
\tag{2}
$$

where $\hat{p}(\boldsymbol{x}, \boldsymbol{y})$ is the empirical distribution of $\{(\boldsymbol{x}_i, \boldsymbol{y}_i)\}_{i=1}^N$, and $\hat{p}(\boldsymbol{x})$ is the empirical distribution of $\{\boldsymbol{x}_i\}_{i=1}^N$. A natural choice of the norm in Equation (1) is a Frobenius norm, computed by

$$
||\nabla_{\boldsymbol{x}} G(\boldsymbol{x}, \boldsymbol{z})|| = \sqrt{\sum_{i=1}^n \sum_{j=1}^m \left[ \frac{\partial G_i(\boldsymbol{x}, \boldsymbol{z})}{\partial x_j} \right]^2}.
$$

## B  Comparison with Related Work

Many papers have denoted to resolving the mode-collapse phenomenon, such as by incorporating divergence measure to reshape the discriminator landscape [8, 9] or generating multi-modal images [19, 20]. A lot of methods focus on the relationship of GANs with changes in the latent noise

or the generator architecture, but the connections with small perturbations on the conditions are relatively less studied. Notably, CcGAN [13] attempts to address the continuous condition issues by adding Gaussian noises to the input conditions. This implies that the model could lose more granular information of the conditions, resulting in outputs that might be less faithful to the input conditions. In particular, when there are large gaps in the dataset, CcGAN must choose large standard deviations for Gaussian noises in order to cover these gaps, which further exacerbates the issue. On the other hand, our proposed method relies on encouraging gradual changes of the output with respect to the input conditions, which does not blur the input conditions themselves.

## C  Connection of Generator Regularization to K-Lipschitz Continuous Conditional Distribution

We formally give the relationship between the conditional distribution learned by a generator and K-Lipschitz continuous conditional distribution aforementioned in Section 3 in Theorem C.1.

**Theorem C.1** *Suppose that the given two arbitrary conditions $\boldsymbol{x}_1$ and $\boldsymbol{x}_2$, the conditional generator $G$ satisfies*

$$||G(\boldsymbol{x}_1, \boldsymbol{z}_0) - G(\boldsymbol{x}_2, \boldsymbol{z}_0)|| \leq K_0 \cdot ||\boldsymbol{x}_1 - \boldsymbol{x}_2||$$

*for any fixed $\boldsymbol{z}_0$. We have*

$$W(G(\boldsymbol{x}_1, \boldsymbol{z}), G(\boldsymbol{x}_2, \boldsymbol{z})) \leq K_0 \cdot ||\boldsymbol{x}_1 - \boldsymbol{x}_2||,$$

*where $\boldsymbol{z} \sim p_{\boldsymbol{z}}(\boldsymbol{z})$.*

We prove Theorem C.1 using the following Lemma C.1.

**Lemma C.1** *Denote the support of a random variable $Z$ as $R_Z$. Functions $f$ and $g$ are defined on $R_Z$. Denote the distribution of $f(Z)$ and $g(Z)$ as $\mathcal{P}_f(Z)$ and $\mathcal{P}_g(Z)$ respectively. If we have $\max_{z \in R_Z} ||f(z) - g(z)|| \leq K$, then the Wasserstein distance between $\mathcal{P}_f(Z)$ and $\mathcal{P}_g(Z)$ satisfies $W(\mathcal{P}_f(Z), \mathcal{P}_g(Z)) \leq K$.*

**Proof C.1** *Denote $X = f(Z)$ and $Y = g(Z)$, and the distribution of $f(Z)$ and $g(Z)$ as $\mathcal{P}_f(Z)$ and $\mathcal{P}_g(Z)$ respectively. Clearly,*

$$\mathcal{P}_f(x) = \int_{z:f(z)=x, z \in R_Z} \mathcal{P}_Z(z) \, dz$$

*and*

$$\mathcal{P}_g(y) = \int_{z:g(z)=y, z \in R_Z} \mathcal{P}_Z(z) \, dz.$$

*The support of $f(Z)$ and $g(Z)$, i.e., the set $\{f(z) : z \in R_Z\}$ and $\{f(z) : z \in R_z\}$ is denoted as $f(R_Z)$ and $g(R_Z)$. Define a joint distribution of $X$ and $Y$ as*

$$\gamma_0(x, y) = \begin{cases} \int_{z:f(z)=x \text{ and } g(z)=y} \mathcal{P}(z) \, dz & z \in R_Z, \text{ s.t. } f(z) = x \text{ and } g(z) = y \\ 0 & o.w. \end{cases} \tag{3}$$

*$\gamma_0$ is intentionally designed such that the marginal distribution of $X$ and $Y$ is precisely $\mathcal{P}_f(X)$ and $\mathcal{P}_g(Y)$:*

$$\int_{x \in f(R_Z)} \gamma_0(x, y) \, dx = \int_{x \in f(R_Z)} \int_{z:f(z)=x \text{ and } g(z)=y} \mathcal{P}_Z(z) \, dz \, dx$$
$$= \int_{z:g(z)=y, z \in R_Z} \mathcal{P}_z(z) \, dz \tag{4}$$
$$= \mathcal{P}_g(y)$$

*and*

$$\int_{y \in g(R_Z)} \gamma_0(x, y) \, dy = \int_{y \in g(R_Z)} \int_{z:f(z)=x \text{ and } g(z)=y} \mathcal{P}_Z(z) \, dz \, dy$$
$$= \int_{z:f(z)=x, z \in R_Z} \mathcal{P}_z(z) \, dz \tag{5}$$
$$= \mathcal{P}_f(x).$$

*By the definition of Wasserstein distance,*

$$
\begin{aligned}
W(\mathcal{P}_f(Z), \mathcal{P}_g(Z)) &= \inf_{\gamma \in \Pi(\mathcal{P}_f, \mathcal{P}_g)} \mathbb{E}_{(x,y) \sim \gamma}[\|x - y\|] \\
&\leq \mathbb{E}_{(x,y) \sim \gamma_0}[\|x - y\|] \\
&= \int_{x \in f(R_Z)} \int_{y \in g(R_Z)} \gamma_0(x, y) \cdot \|x - y\| \, dx \, dy \\
&= \int_{x \in f(R_Z)} \int_{y \in g(R_Z)} \int_{z: f(z)=x \text{ and } g(z)=y} \mathcal{P}(z) \cdot \|x - y\| \, dz \, dx \, dy \\
&= \int_{z \in R_Z} \mathcal{P}_z(z) \cdot \|f(z) - g(z)\| \, dz \\
&\leq \int_{z \in R_Z} \mathcal{P}_Z(z) \cdot K \, dz \\
&= K.
\end{aligned}
$$

The proof of Theorem C.1 is obvious using Lemma C.1. In Theorem C.1, given the fixed $\boldsymbol{x}_1$ (or $\boldsymbol{x}_2$), the generator $G$ can be viewed as a function $G(\boldsymbol{x}_1, \cdot)$ (or $G(\boldsymbol{x}_2, \cdot)$) that maps a random noise $\boldsymbol{z}$ to $G(\boldsymbol{x}_1, \boldsymbol{z})$ (or $G(\boldsymbol{x}_2, \boldsymbol{z})$). Take the random variable $Z$ in Lemma C.1 as $\boldsymbol{z}$. Take $G(\boldsymbol{x}_1, \cdot)$ and $G(\boldsymbol{x}_2, \cdot)$ as the functions $f$ and $g$ in Lemma C.1. Then Theorem C.1 is evident.

## D    Algorithm for GR-cGAN Training

We give the algorithms for training a GR-cGAN. If the generator regularization takes the form in Equation 1, an algorithm for training a GR-cGAN is given in Algorithm 1.

The direct evaluation of Equation (1) is computationally prohibitive when the dimensions $m$ and $n$ are high. When the dimension of the condition and the dimension of generator output are high (say, more than 100), we provide an alternative to Equation (1) by locally approximating the gradient in a finite difference fashion:

$$
\mathcal{L}_{\widetilde{GR}}(G) \quad = \mathbb{E}_{\substack{\boldsymbol{z} \sim p_z(\boldsymbol{z}), \\ \boldsymbol{x} \sim \tilde{p}(\boldsymbol{x})}} [\min(f(\boldsymbol{x}, \Delta\boldsymbol{x}, \boldsymbol{z}), \tau_1)] \tag{6}
$$

where

$$
f(\boldsymbol{x}, \Delta\boldsymbol{x}, \boldsymbol{z}) = \frac{\|G(\boldsymbol{x} + \Delta\boldsymbol{x}, \boldsymbol{z}) - G(\boldsymbol{x}, \boldsymbol{z})\|}{\|\Delta\boldsymbol{x}\|},
$$

$\Delta\boldsymbol{x} \sim p_{\Delta\boldsymbol{x}}(\Delta\boldsymbol{x})$ is a small perturbation added to $\boldsymbol{x}$ and $p_{\Delta\boldsymbol{x}}(\Delta\boldsymbol{x})$ is the distribution of $\Delta\boldsymbol{x}$. The distribution $p_{\Delta\boldsymbol{x}}(\Delta\boldsymbol{x})$ is designed to be a distribution centered close to zero and has a small variance, such as a normal distribution. $\tau_1$ is a bound for ensuring numerical stability. We also impose a lower bound $\tau_2$ on $\Delta\boldsymbol{x}$ for the same reason. If the generator regularization takes the approximated form in Equation 6, the training algorithm is given in Algorithm 2.

## E    More Details and Results of the Experiments

### E.1    Circular 2-D Gaussians

#### E.1.1    Full Dataset

The positions of the train labels are shown in Figure 6(a). To generate a training set, for each x in the train labels, 10 samples are generated. Figure 6(b) shows 1,200 training samples. All considered models are trained on the same training set for 6,000 iterations. We use the same value of $R$ and $\tilde{\sigma}^2$ as in Section 4.1.1.

**Evaluation metrics and quantitative results.** We choose 360 values of $\boldsymbol{x}$ evenly from the interval $[0, 2\pi]$. For each model, given a value of $\boldsymbol{x}$, we generate 100 samples, yielding 36,000 fake samples in total. A circle with $(\sin(\boldsymbol{x}), \cos(\boldsymbol{x}))$ as the center and $2.15\tilde{\sigma}$ as the radius can enclose about 90% of the volume inside the pdf of $\mathcal{N}(\boldsymbol{\mu}_{\boldsymbol{x}}, \boldsymbol{\Sigma})$. We define a fake sample $\boldsymbol{y}$ as a high quality sample if its Euclidean distance from $\boldsymbol{y}$ to $(\sin(\boldsymbol{x}), \cos(\boldsymbol{x}))$ is smaller than $2.15\tilde{\sigma} = 0.43$. A mode (i.e., a

---

**Algorithm 1** An algorithm for training GR-cGAN with generator regularization as in Equation 1

---

**Require:** The generator regularization coefficient $\lambda$, the training set $\{\boldsymbol{x}_i, \boldsymbol{y}_i\}_{i=1}^N$, the batch size $m$, the number of iterations of the discriminator per generator iteration $n$, Adam hyper-parameters $\alpha$, $\beta_1$ and $\beta_2$, the number of iterations $K$.
**Require:** $w_0$, initial discriminator parameters. $\theta_0$, initial generator's parameters.

1: **for** $k = 1$ **to** $K$ **do**
2:    **for** $t = 1, \ldots, n$ **do**
3:       Sample a batch of real samples from the training set, denote as $\{\boldsymbol{x}_j, \boldsymbol{y}_j\}_{j=1}^m$.
4:       Sample a batch of random noises independently, $\boldsymbol{z}_j \sim p_z(\boldsymbol{z})$, for $j = 1, 2, \ldots, m$.
5:       Discriminator loss $\leftarrow \frac{1}{m}\sum_{j=1}^m \left[\log D(\boldsymbol{x}_j, \boldsymbol{y}_j) + \log(1 - D(G(\boldsymbol{x}_j, \boldsymbol{z}_j)))\right]$
6:       Update $D$.
7:    **end for**
8:    Sample two batches of real samples from the training set independently, denote as $\{\boldsymbol{x}_j, \boldsymbol{y}_j\}_{j=1}^m$ and $\{\boldsymbol{x}_j', \boldsymbol{y}_j'\}_{j=1}^m$.
9:    Sample a batch of random noises independently, $\boldsymbol{z}_j \sim p_z(\boldsymbol{z})$ for $j = 1, 2, \ldots, m$.
10:   Sample random numbers $\epsilon_j \sim U[0, 1]$ for $j = 1, 2, \ldots, m$.
11:   $\boldsymbol{x}_j'' \leftarrow \epsilon\boldsymbol{x}_j + (1 - \epsilon)\boldsymbol{x}_j'$ for $j = 1, 2, \ldots, m$.
12:   $\mathcal{L}_{GR}(G) \leftarrow \frac{1}{m}\sum_{j=1}^m ||\nabla_{\boldsymbol{x}_j''} G(\boldsymbol{x}_j'', \boldsymbol{z}_j)||$
13:   Generator loss $\leftarrow \frac{1}{m}\sum_{j=1}^m[\log(1 - D(G(\boldsymbol{x}_j, \boldsymbol{z}_j)))] + \lambda\mathcal{L}_{GR}(G)$
14:   Update $G$.
15: **end for**

---

---

**Algorithm 2** An algorithm for training GR-cGAN with generator regularization as in Equation 6

---

**Require:** The generator regularization coefficient $\lambda$, the training set $\{\boldsymbol{x}_i, \boldsymbol{y}_i\}_{i=1}^N$, the batch size $m$, the number of iterations of the discriminator per generator iteration $n$, Adam hyper-parameters $\alpha$, $\beta_1$ and $\beta_2$, the number of iterations $K$.
**Require:** $w_0$, initial discriminator parameters. $\theta_0$, initial generator's parameters.

1: **for** $k = 1$ **to** $K$ **do**
2:    **for** $t = 1, \ldots, n$ **do**
3:       Sample a batch of real samples from the training set, denote as $\{\boldsymbol{x}_j, \boldsymbol{y}_j\}_{j=1}^m$.
4:       Sample a batch of random noises independently, $\boldsymbol{z}_j \sim p_z(\boldsymbol{z})$, for $j = 1, 2, \ldots, m$.
5:       Discriminator loss $\leftarrow \frac{1}{m}\sum_{j=1}^m \left[\log D(\boldsymbol{x}_j, \boldsymbol{y}_j) + \log(1 - D(G(\boldsymbol{x}_j, \boldsymbol{z}_j)))\right]$
6:       Update $D$.
7:    **end for**
8:    Sample two batches of real samples from the training set independently, denote as $\{\boldsymbol{x}_j, \boldsymbol{y}_j\}_{j=1}^m$ and $\{\boldsymbol{x}_j', \boldsymbol{y}_j'\}_{j=1}^m$.
9:    Sample a batch of random noises independently, $\boldsymbol{z}_j \sim p_z(\boldsymbol{z})$ for $j = 1, 2, \ldots, m$.
10:   Sample random numbers $\epsilon_j \sim U[0, 1]$ for $j = 1, 2, \ldots, m$.
11:   $\boldsymbol{x}_j'' \leftarrow \epsilon\boldsymbol{x}_j + (1 - \epsilon)\boldsymbol{x}_j'$ for $j = 1, 2, \ldots, m$.
12:   Sample a batch of perturbations $\Delta\boldsymbol{x}_j \sim p_{\Delta\boldsymbol{x}}(\Delta\boldsymbol{x})$ for $j = 1, 2, \ldots, m$
13:   $\mathcal{L}_{\widetilde{GR}}(G) \leftarrow \frac{1}{m}\sum_{j=1}^m[\min(f(\boldsymbol{x}_j'', \Delta\boldsymbol{x}_j, \boldsymbol{z}_j), \tau_1)]$, where $f(\boldsymbol{x}_j'', \Delta\boldsymbol{x}_j, \boldsymbol{z}_j) = \frac{\|G(\boldsymbol{x}_j'' + \Delta\boldsymbol{x}_j, \boldsymbol{z}_j) - G(\boldsymbol{x}_j'', \boldsymbol{z}_j)\|}{\|\Delta\boldsymbol{x}_j\|}$.
14:   Generator loss $\leftarrow \frac{1}{m}\sum_{j=1}^m[\log(1 - D(G(\boldsymbol{x}_j, \boldsymbol{z}_j)))] + \lambda\mathcal{L}_{\widetilde{GR}}(G)$
15:   Update $G$.
16: **end for**

---

Gaussian) is recovered if at least one high quality sample is generated. For the conditional distribution given by the generator, (i.e., the distribution of $G(\boldsymbol{x}, \boldsymbol{z})$ with $\boldsymbol{z} \sim p_z(\boldsymbol{z})$), we assume this distribution is Gaussian and estimate its mean and covariance using 100 fake samples, denoted by $\boldsymbol{\mu}_{\boldsymbol{x}}^G$ and $\boldsymbol{\Sigma}_{\boldsymbol{x}}^G$ respectively. We compute the **2-Wasserstein Distance (W2)** [21] between the true conditional distribution and the distribution given by the generator, in other words, the 2-Wasserstein Distance

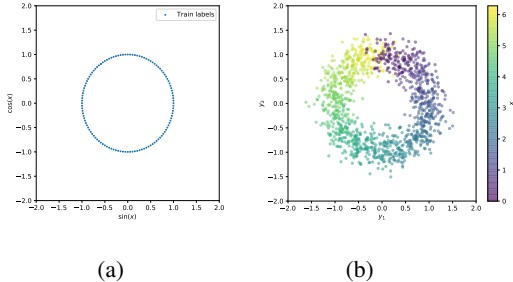

(a)                                        (b)

Figure 6: (a) plots the locations of the means of the 120 Gaussians. (b) illustrates 1,200 randomly chosen samples from the training set.

between

$$\mathcal{N}\left(\left(\begin{array}{c} R \cdot \sin(\boldsymbol{x}) \\ R \cdot \cos(\boldsymbol{x}) \end{array}\right), \tilde{\sigma}^2 I_{2\times 2}\right) \text{ and } \mathcal{N}(\boldsymbol{\mu}_{\boldsymbol{x}}^G, \boldsymbol{\Sigma}_{\boldsymbol{x}}^G).$$

The whole experiment is repeated three times and the averaged values of the metrics are reported in Table 2 over three repetitions. We see that GR-cGAN demonstrates competitive performance against CcGAN, especially in terms of the 2-Wasserstein distance.

| Model | % High Quality | % Recovered Mode | 2-Wasserstein Dist. |
|---|---|---|---|
| CcGAN (HVDL) | **95.9** | 100 | $3.79 \times 10^{-2}$ |
| CcGAN (SVDL) | 91.8 | 100 | $5.37 \times 10^{-2}$ |
| Deg. GR-cGAN | **95.9** | 100 | $3.79 \times 10^{-2}$ |
| GR-cGAN | 93.7 | **100** | $\mathbf{2.63 \times 10^{-2}}$ |

Table 2: Evaluation metrics for the full dataset experiments. The metrics of 36,000 fake samples generated from each model over three repetitions are given. Larger values of "% Recovered Mode" are better, while smaller values of "2-Wasserstein Dist." are preferred. Note that the larger values of "% High Quality." does not completely mean that the GAN model is better, because the samples generated by a GAN whose distribution is concentrated to a point located within the threshold will also be considered as high-quality. We use these evaluation metrics since they are also adopted in [13].

**Visual results.** We select 8 angles that do not exist in the training set. For each angles $\boldsymbol{x}$ selected, we use all the models to generate 100 fake samples. Furthermore, we plot the circle with $(\sin(\boldsymbol{x}), \cos(\boldsymbol{x}))$ as the center and $2.15\tilde{\sigma}$ as the radius to indicate the true conditional distribution $\mathcal{N}(\boldsymbol{\mu}_{\boldsymbol{x}}, \boldsymbol{\Sigma})$. The results are given in Figure 7. Fake samples from our method better match the true samples when compared to the other methods.

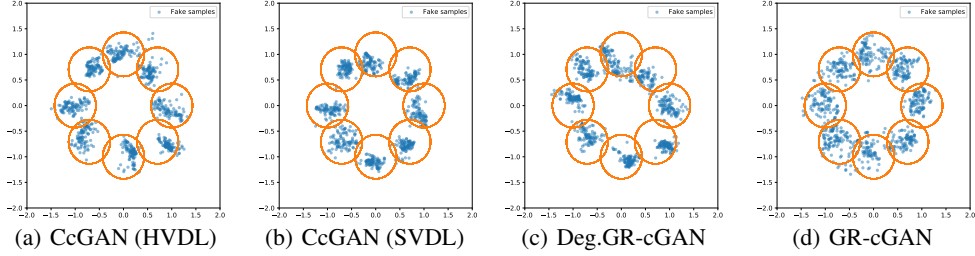

(a) CcGAN (HVDL)      (b) CcGAN (SVDL)      (c) Deg.GR-cGAN      (d) GR-cGAN

Figure 7: Visual results of the Circular 2-D Gaussians experiments on the full dataset. For each subfigure, we generate 100 fake samples using each model at each of the 8 means that are absent from the training set. The blue dots represent the fake samples. For each mean $\boldsymbol{x}$ given, the circle locates at $(\sin(\boldsymbol{x}), \cos(\boldsymbol{x}))$ and has a radius of $2.15\tilde{\sigma}$, which can cover about 90% of the volume inside the pdf of $\mathcal{N}(\boldsymbol{\mu}_{\boldsymbol{x}}, \boldsymbol{\Sigma})$.

### E.1.2 Partial Dataset

We provide the quantitative evaluation metrics for the experiment in Section 4.1, as shown in Table 3.

| Model | % High Quality | % Recovered Mode | 2-Wasserstein Dist. |
|---|---|---|---|
| CcGAN (HVDL) | 91.0 | 100 | $3.77 \times 10^{-2}$ |
| CcGAN (SVDL) | **95.7** | 100 | $3.59 \times 10^{-2}$ |
| Deg. GR-cGAN | 93.9 | 100 | $4.51 \times 10^{-2}$ |
| GR-cGAN | 93.5 | **100** | $\mathbf{3.06 \times 10^{-2}}$ |

Table 3: Evaluation metrics for partial dataset experiments. The details of these metrics can be found in Appendix E.1.1. The metrics of 36,000 fake samples generated from each model over three repetitions are given.

### E.1.3 Training Details

**Network architectures:** We use the same network architecture setting as in [13]. Please refer to Table 4 for details.

| (a) Generator |
|---|
| $\boldsymbol{z} \in \mathbb{R}^2 \sim N(0, I); \boldsymbol{y} \in \mathbb{R}$ |
| $\mathrm{concat}(\boldsymbol{z}, \sin(\boldsymbol{y}), \cos(\boldsymbol{y})) \in \mathbb{R}^4$ |
| $\mathrm{fc} \to 100; \mathrm{BN}; \mathrm{ReLU}$ |
| $\mathrm{fc} \to 100; \mathrm{BN}; \mathrm{ReLU}$ |
| $\mathrm{fc} \to 100; \mathrm{BN}; \mathrm{ReLU}$ |
| $\mathrm{fc} \to 100; \mathrm{BN}; \mathrm{ReLU}$ |
| $\mathrm{fc} \to 100; \mathrm{BN}; \mathrm{ReLU}$ |
| $\mathrm{fc} \to 100; \mathrm{BN}; \mathrm{ReLU}$ |
| $\mathrm{fc} \to 2$ |

| (b) Discriminator |
|---|
| A sample $\boldsymbol{x} \in \mathbb{R}^2$ with label $\boldsymbol{y} \in \mathbb{R}$ |
| $\mathrm{concat}(x, \sin(y), \cos(y)) \in \mathbb{R}^4$ |
| $\mathrm{fc} \to 100; \mathrm{ReLU}$ |
| $\mathrm{fc} \to 100; \mathrm{ReLU}$ |
| $\mathrm{fc} \to 100; \mathrm{ReLU}$ |
| $\mathrm{fc} \to 100; \mathrm{ReLU}$ |
| $\mathrm{fc} \to 100; \mathrm{ReLU}$ |
| $\mathrm{fc} \to 1; \mathrm{Sigmoid}$ |

Table 4: Network architectures for the generator and the discriminator of the experiments in Section 4.1. "fc" represents a fully-connected layer. "BN" denotes batch normalization. The label $\boldsymbol{y}$ is treated as a real scalar so its dimension is 1. We use $\boldsymbol{y}$, $\sin(\boldsymbol{y})$ and $\cos(\boldsymbol{y})$ together as the input to the generator networks.

**Training steps:** The training steps is also the same as in [13]. All GANs are trained for 6000 iterations on the training set with the Adam (Kingma & Ba, 2015 ) optimizer (with $\beta_1 = 0.5$ and $\beta_2 = 0.999$ ), a constant learning rate of $5 \times 10^{-5}$ and a batch size of 128. The hyperparameters of CcGAN takes the same value as in Section S.VI.B of [13]. The $\lambda$ of GR-cGAN is set to 0.02, with the generator regularization term computed by Equation 1.

### E.2 Conditional Time Series Generation Experiments

### E.2.1 Evaluation Details

In TimeGAN, the datasets are only split into train-test during evaluation. However, when deployed to the real world, the generation models must be robust to conditions that do not appear in training. In light of this, we first partition the real dataset into train and test before GAN training, where the $30\%$ test set is not exposed during generator training. To compute the predictive score, we first partition the generated dataset into train-validation and train until $50,000$ iterations or an early stopping criteria of $3,000$ iterations is met. The best set of weights is used to forecast on the entire original dataset. As for the discriminative score, we further partition both the generated and the real datasets into train-valid-test by $80\% - 10\% - 10\%$. A shorter early stopping criteria of $1,500$ iterations is used due to its longer training time. For both predictive and discriminative scores, we train the forecast model and the discriminative model using 10 different seeds and data splits.

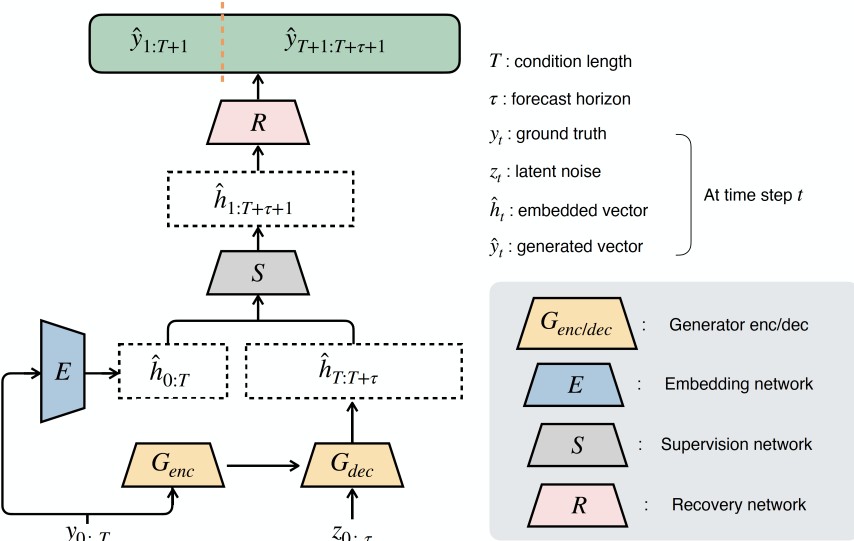

Figure 8: The modified TimeGAN architecture in order to support conditional inputs, which we denote as **cTimeGAN**. Note that $\hat{h}_{0:T}$ is not generated by $G_{\text{enc}}$, as we can directly leverage the Embedding network ($E$) trained via direct supervision in an autoencoder manner.

In contrast to TimeGAN which uses a simple GRU with minimal hidden units, we argue that it is essential to use a more powerful model for predictive score. Otherwise, even if the generation model was able to generate complex real-world dynamics, a simple forecasting model could easily fail to capture it. We also report the predictive score under multi-step forecasting, where the forecast horizon is always set to the last one-third of the window. On the other hand, a simpler model is desired for discriminative score, as the task of discriminator is considered to be much simpler than generation. Lastly, to highlight the robustness of our model to unseen results, we also report the results when the test set is included in the training of the generator under the dataset name ETTm1$_{\text{all}}$.

As for the discriminative score model, we use a two-layer GRU with 5 hidden units and a dropout of 0.1. For the forecasting model, we use a two-layer encoder-decoder transformer model with sinusoidal positional attention. $d\_\text{model} = 20, d_{ff} = 30$, num_heads $= 2$. Dropout is also 0.1. We notice that the datasets could be small if we train the forecasting model only using generated data of the same size as the original data. Thus, we let the generated data to be 10 times the size of the original data for ETTm1 and 100 times for stock. Even so, the amount of patterns in the two datasets are still limited. Therefore, we also randomly mask out[22] 10% of the data during training for ETTm1 and 20% of the data for stock in order to obtain more stable forecasting performance.

### E.2.2 Model Details

Note that the output of the $G$-encoder is not used directly to generate the hidden states $[\hat{h}_{0:T}]$, which are instead generated by directly employing the $E$ module. This leads to better results in our experiments, as we hypothesize that mapping $y$ to $\hat{h}$ is considerably different from extracting meaningful information to assist the mapping from $z$ to $\hat{h}$. The rest of the architecture remains the same as TimeGAN's.

Before training GR-cGAN, we first train a VAE to map a pair of conditions $\left([y_{1,i}]_{i=0}^{T}, [y_{2,i}]_{i=0}^{T}\right)$ into a latent space with a prior of $\mathcal{N}(0, \mathcal{I})$ to get $(\mu_1, \mu_2)$. Then, we randomly sample $\alpha \in (0, 1)$ and feed $\tilde{\mu} = \alpha\mu_1 + (1 - \alpha)\mu_2$ back into the VAE decoder to get the reconstructed conditions $\left([\tilde{y}_{\tilde{\mu},i}]_{i=0}^{T}, [\tilde{y}_{\tilde{\mu}+\epsilon,i}]_{i=0}^{T}\right)$. At the end, our gradient penalty becomes $\mathcal{L}_{GR}(G) = ||G([\tilde{y}_{\tilde{\mu},i}]_{i=0}^{T}, [z_{T+i}]_{i=1}^{\tau}) - G([\tilde{y}_{\tilde{\mu}+\epsilon,i}]_{i=0}^{T}, [z_{T+i}]_{i=1}^{\tau})||$. Note that we omit $\sigma_1, \sigma_2$ from the VAE encoder for simplicity, but noticed little impact on performance.

We use three-layer GRUs for all the submodules in cTimeGAN. For ETTm1, the hidden dimension is set to 30, $z$ dimension is 12, and dropout is 0.1. For stock, the hidden dimension is set to 24, $z$

dimension is 24, and dropout is 0. The rest of the hyperparameters stay the same as TimeGAN's. As for the VAE module, we adapted a simple implementation from Github[6] and replaced its 2D-convolutional layers with stacked 1D-convolutional layers. To improve its stability, we further use a cyclical annealing schedule [23] as well as a discriminator to facilitate more realistic reconstructions. KL-regularization weight is always set to $0.0001$. All of our hyperparameters can be found in the released code. All the hyperparameters except $\lambda$ are chosen heuristically.

## E.3  Sensitivity Analysis

We first show in Figure 5 and Figure 9 the sensitivity of our model under different generator penalty weights $\lambda$. For CcGAN, when $\epsilon = 0.1$, discriminative = 0.421 and predictive = 0.071. When $\epsilon = 0.01$, discriminative = 0.468 and predictive = 0.052. When $\epsilon = 0.001$, discriminative = 0.390, predictive = 0.0599.

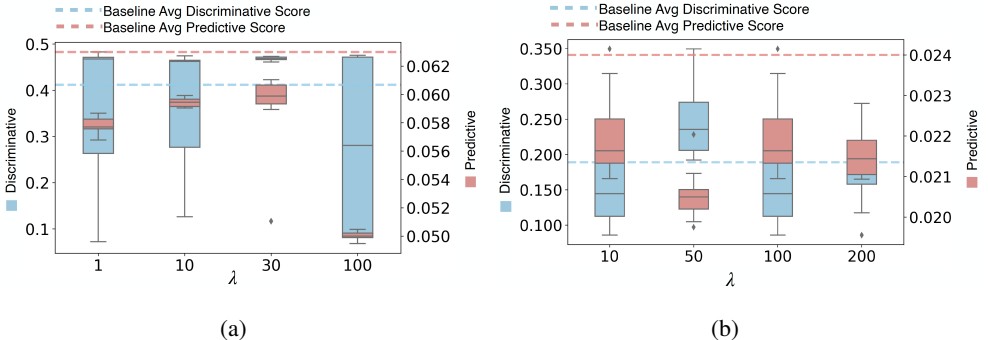

(a)                                           (b)

Figure 9: Sensitivity analysis of the generator regularizor weight $\lambda$ on (a) ETTm1. (b) stock. The horizontal dashed lines are the discriminative scores (blue) and predictive scores (orange) for the baseline cTimeGAN. The confidence intervals are constructed by evaluating the generated data 10 times. Lower scores indicate better results. For example, on the left-hand side, we see that the red box corresponding to $\lambda = 30$ falls entirely under the red dashed line. This means that in all ten runs our model surpasses the averaged baseline result. Meanwhile, we also note that there are large variances in the discriminative score. We leave the redesign of the discriminative score evaluation framework to future studies.

### E.3.1  Case Studies

---

[6]https://github.com/AntixK/PyTorch-VAE.

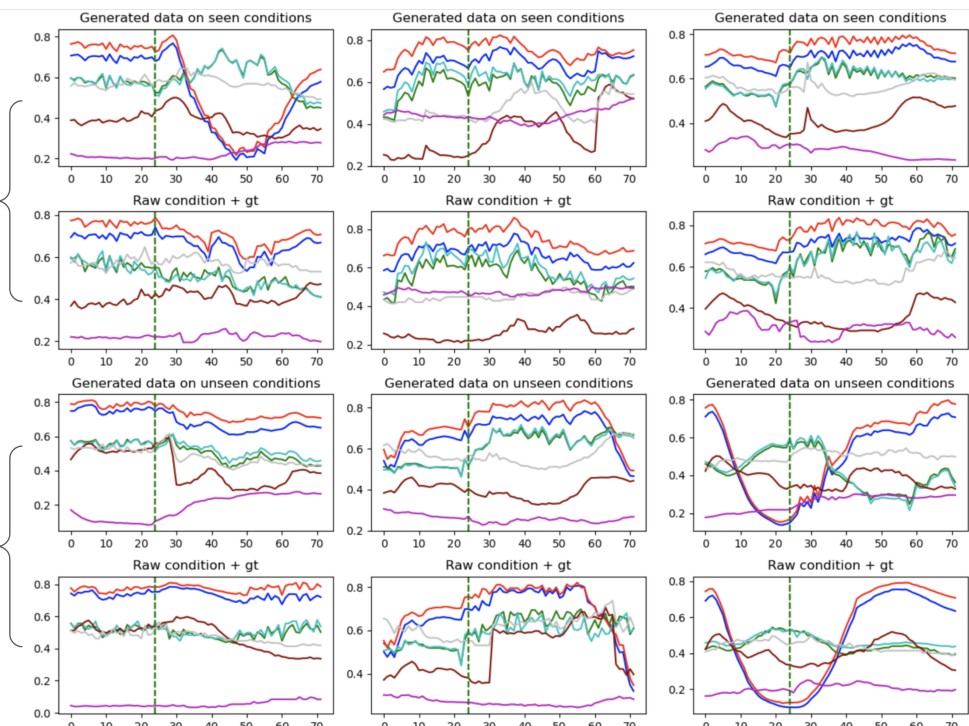

Figure 10: Our model with $\lambda = 100$ on ETTm1. The vertical green dashed lines separate the conditions from the generated results. The first two rows are the generation results on conditions seen during training, where the first row shows the generated data and the corresponding figures in the second row show the corresponding original data (e.g. The image in the first column and the first row uses the same condition as the image in the first column and the second row, where the former is the generated window and the latter is the ground truth). The last two rows are the generation results on conditions not seen during training. Our model is still able to capture the general trend and output realistic samples.

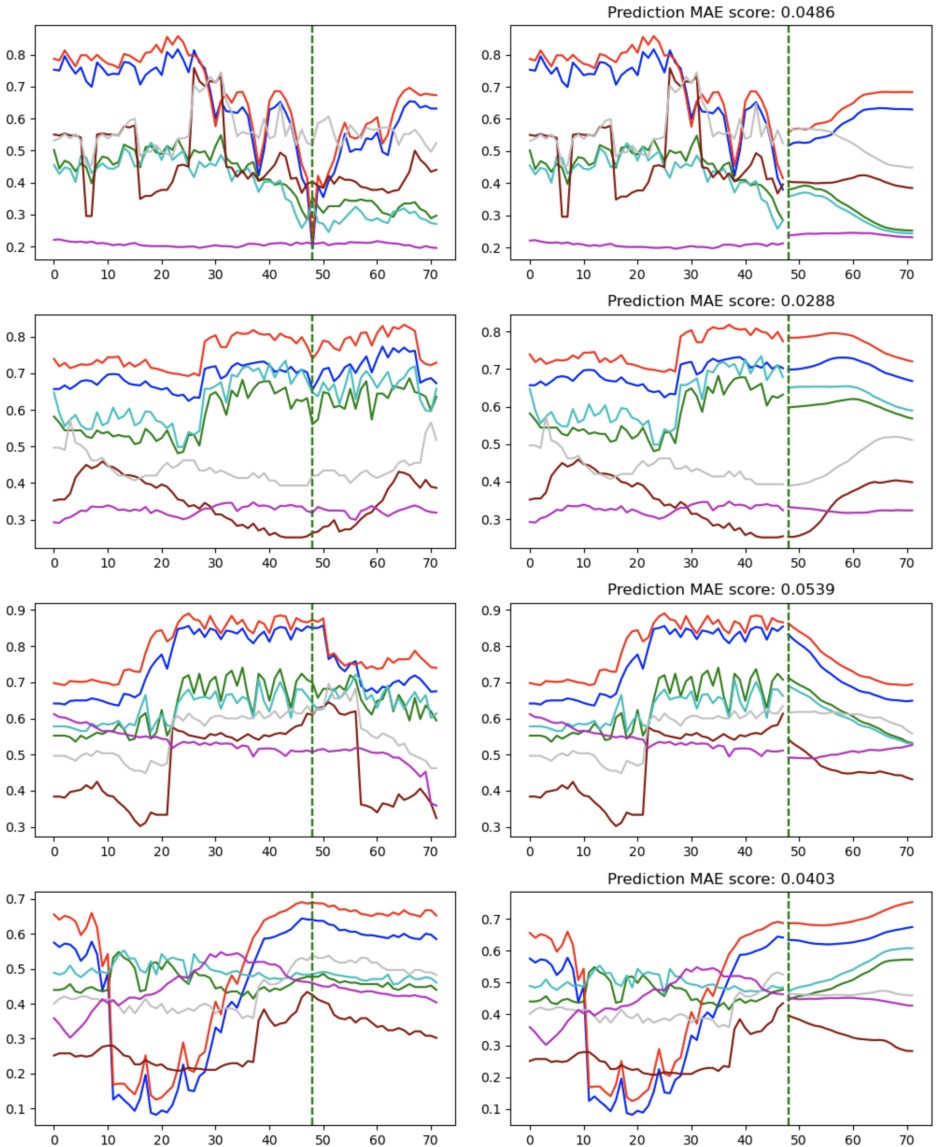

Figure 11: A forecasting model trained with the data generated by our model with $\lambda = 100$ on ETTm1. The forecasting model is able to predict the overall trend, indicating that the generated data has properties that are faithful to the original data.

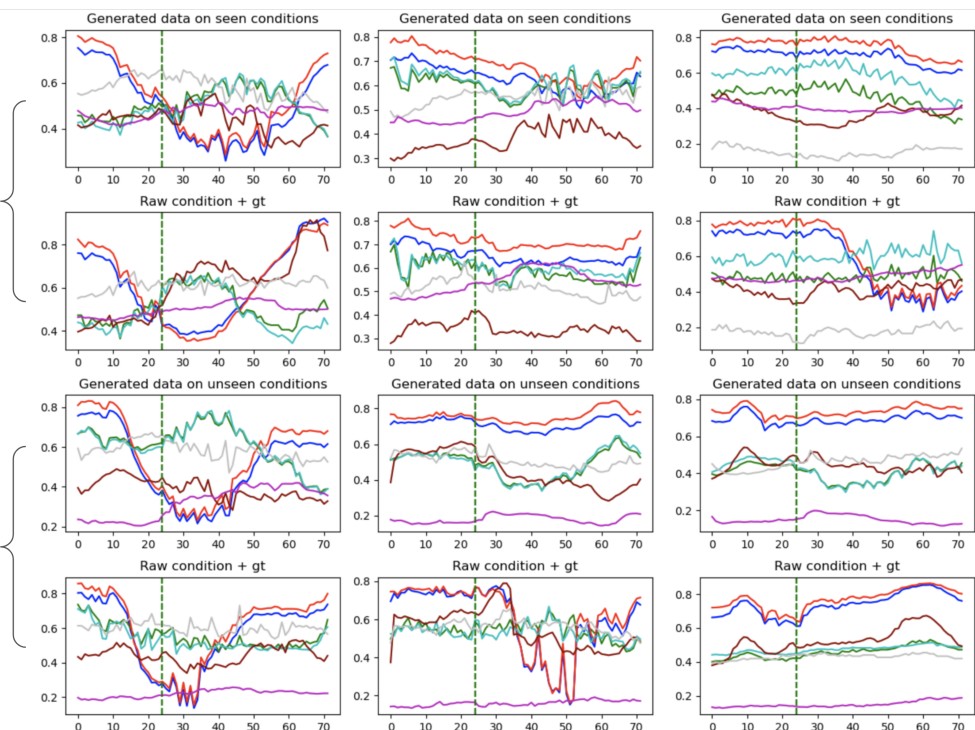

Figure 12: The baseline model cTimeGAN on ETTm1. The vertical green dashed lines separate the conditions from the generated results. The first two rows are the generation results on conditions seen during training, where the first row shows the generated data and the corresponding figures in the second row show the corresponding original data (e.g. The image in the first column and the first row uses the same condition as the image in the first column and the second row, where the former is the generated window and the latter is the ground truth). The last two rows are the generation results on conditions not seen during training. The model behaves considerably worse on unseen conditions.

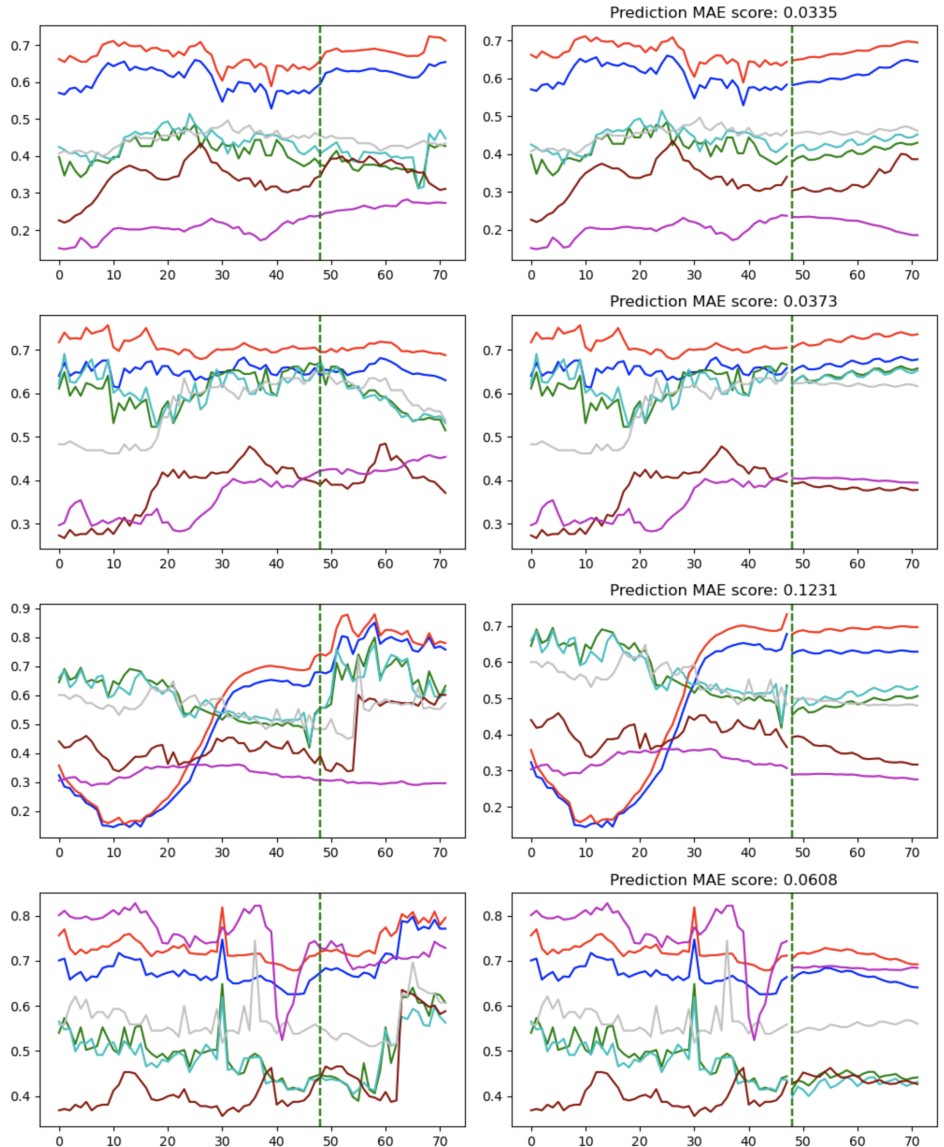

Figure 13: A forecasting model trained with the data generated by cTimeGAN on ETTm1. The forecasting model tends to give roughly flat predictions, indicating that the generated data is less consistent with the original version.

