# OpenReview forum: "Mind Your Step: Continuous Conditional GANs with Generator Regularization"
_NeurIPS.cc/2022/Workshop/SyntheticData4ML — Neurips 2022 SyntheticData4ML_

### Official Review · Reviewer_rJUQ · 2022-10-11
**A regularization method in the form of Lipschitz penalty, but not good enough**

**Rating:** 3
**Confidence:** 4

**Review:**

This paper focus on two challenges for continuous conditional GAN. To solve these two challenges, the authors proposed a regularization term of norm of Jacobian matrix and uses some experiments to show the performance of their regularization method.


## Strength
The experiments are well-developed. This regularization method shows good performance in the experiment part under the metric the authors used.


## Weakness
This work is lack of novelty. Regularization methods about Jacobian matrix has been proposed years ago in other models. I suggest that the authors mention more existing regularization methods or relative methods, and compare existing ones with the authors' method. What's the difference between the authors' method and existing methods? Instead of using on GAN, I can't tell the difference.


I think the authors need to explain more about how this regularization method works for the challenges. The author mentioned that *it is natural to assume that a minor perturbation to the condition x will only slightly disturb the conditional distribution*, this follows the definition of continuity. However, when regularizing the norm of Jacobian matrix, it gives a situation similar with Lipchitz continuity. The authors also defined a concept of Lipschitz Continuous Conditional Distribution. Then why it is reasonable to consider Lipchitz continuity? The author didn't explain why use this stronger assumption. The authors want a smooth distribution, but does it work for real data? I think it could be better to experiment to show real work data follows Lipchitz continuity or some properties similar with Lipchitz continuity. That could be more convincing. The second challenge the authors mentioned also need more explanation. Under what situation, people will increase the dimension number? How *conditions observed become more sparse* happens? Do the authors means on the same dataset or between different datasets? Does the sparse situation happens in the real datasets, especially in the dataset used in experiment?


The authors didn't mention the time complexity in the algorithm or training cost in the experiment. If it is too slow with this regularization method, then no one will use it.


The paper’s writing needs a lot of improvement. For example, the sentences should read correctly if the citation is delete.


## Summary:
Good and simple idea, but lack of novelty. This paper need more refinement. The motivation also need more explanation.

---

### Official Review · Reviewer_B4gv · 2022-10-18
**Interesting ideas with good results**

**Rating:** 7
**Confidence:** 4

**Review:**

This paper proposes a simple generator regularization term on the GAN generator loss in the form of a Lipschitz penalty. The intuition of this Lipszhitz penalty is that, when the generator is fed with neighboring conditions in the continuous space, the regularization term will leverage the neighbor information and push the generator to generate samples that have similar conditional distributions for neighboring conditions. This paper also analyzes the effect of the proposed regularization term and demonstrate its robust performance on a range of synthetic tasks as well as real-world conditional time series generation tasks.

This paper presents interesting ideas and promising results.

---

### Meta-Review · Area_Chair_1haz · 2022-10-20

**Recommendation:** Accept

**Review:**

The motivation for the regularization term in GAN training is clear, however, the two reviewers disagree on the novelty, with rJUQ arguing that the penalty on the Jacobian has limited novelty and potentially computational challenges. While I agree with this assessment, the work also represents a step in the right direction for improving continuous conditional GANs and has promising experimental results.

As such, it should be of interest to the community. I am recommending acceptance but I hope the authors can take into account the feedback of both reviewers to improve the paper.